# Temporary Disability Pension, RTW-Intentions, and RTW-Behavior: Expectations and Experiences of Disability Pensioners over 17 Months

**DOI:** 10.3390/ijerph17010238

**Published:** 2019-12-28

**Authors:** Sonia Lippke, Natalie Schüz, Elisabeth Zschucke

**Affiliations:** 1Department of Psychology & Methods, Jacobs University Bremen, 28759 Bremen, Germany; 2Deutsche Rentenversicherung Oldenburg-Bremen, 28209 Bremen, Germany

**Keywords:** return to work, intention, self-efficacy, outcome expectancies, medical rehabilitation

## Abstract

*Purpose*: Individuals receiving a temporary disability pension (TDP) should get the opportunity to return to work (RTW). The current study aims to determine the factors contributing to RTW. *Methods*: 453 individuals on TDP were interviewed at three measurement points (T1, T2 = T1 + approximately 7 months, T3 = T1 + approximately 17 months). Socio-demographics, psychological predictors, and current work status (maintained TDP, permanent disability pension, old-age pension, employed, or receipt of other benefits) were assessed. *Results*: Throughout the duration of the study, only four of the former temporary disability pensioners returned to work, and an additional seven made themselves available to the labor market. These were individuals who were younger, in TDP for a shorter period of time, and reported a higher RTW-intention. Higher RTW-intention was cross-sectionally associated with younger age, shorter TDP duration, and more positive outcome expectancies. Additionally, study participants who expected that medical rehabilitation would help them RTW were more motivated to RTW. An increase in RTW-intention over 17 months was related to younger age and better health. *Conclusion*: Personal factors such as self-efficacy and job-related variables appear less important for RTW than age and subjective health status. The observed RTW rates call for early support and tailored medical rehabilitation interventions that help individuals prevent functional limitations, overcome disability, and facilitate RTW.

## 1. Introduction

A better understanding of the complex processes underlying return to work (RTW), such as factors relating to aging, impairment, disability, and subjective health, is required. Numerous studies have investigated the risk of long-term sick leave and entering *disability pension* (DP) [1,2,3,4,5,6,7]. However, there are only a few studies investigating RTW after receiving DP and the various factors facilitating RTW. This dearth of studies may partially be due to the fact that this is a difficult population to study—RTW rates are usually quite low and disability pensioners may feel stigmatized and reluctant to talk about their status. One of the few intervention studies involving disability pensioners did not find any significant effects on RTW: After one year, participants in the intervention group were not significantly more likely to RTW or be seeking a job than participants in the control group. However, the lack of significant findings may be due to a small sample size (*N* = 89) or a low response rate, as the return rate for the intervention group (22%) was twice as high as the control group (11%). The authors also found that positive expectations, better physical functioning, and health made RTW more likely [8]. In general, small sample sizes and low response rates are not only a problem but also indicative of the challenges experienced by individuals facing long-term disabilities: Preclusion from work and employment in specific, and participation in society in general—which are human rights [9].

*Medical rehabilitation* is a key instrument for improving long-term health, functioning, and social participation [10,11,12]. The role of medical rehabilitation in relation to other factors is still under-researched in the population of temporary disability pensioners [6,12]. *Intention*, which is also referred to as motivation, goal, commitment, or decision-making, is another important factor in this context [11,13]. Intention is specifically important for behavior adoption [11,14,15]. Furthermore, prior employment status emerged as important, and prior work experience with potential occupational health promotion is necessary for successful RTW and according intentions [16]. In general, studies have shown that successful RTW is related to prior intentions [6,8,15,17,18,19], and to social support [13,19]. The *Job Demands-Resources Model* [20] looks at work-related intention. In this model, intention is determined by work-related resources (*Job Resources*) and personal resources (*Personal Resources*), along with work-related challenges (*Job Demands*) [20]. Accordingly, Nigatu and colleagues [5] found that too many work-related challenges hinder work ability and RTW. This finding was also replicated by Haveraaen and colleagues [21], who found that *decision-making*, *self-efficacy expectations*, and *social support* are important for RTW.

Various authors have pointed out that interventions should come to pass as early as possible [6,22,23]. Overland and colleagues [1] found that age was not correlated with RTW-intentions but that in general, RTW was associated with age [12]. Additionally, *outcome expectancies* (OE) related to incentives or gratification of work also appear to be important in RTW [24]. Moreover, individuals who managed to RTW were more likely to be *previously employed* [16,17,25] and had participated in a specialized rehabilitation during their DP [25]. In summation, the following *research questions* remain unanswered and the main aim of this work is to investigate them accordingly. The research questions are as follows:(1)What is the likelihood of returning to work (RTW) when receiving temporary disability pension (TDP) in Germany, and what are the characteristics of those who returned to work?(2)What is the relationship between RTW-behavior and RTW-intention?(3)Is there an association between the duration of TDP and RTW-intention?(4)What socio-demographic and psychological factors predict RTW-intention?

## 2. Materials and Methods

Analyses of the baseline data (T1) of this study have been published elsewhere [26], but the follow-up measurement points have not been published. Like T1, the second and third measurement points were assessed via computer-assisted telephone interviews (CATI, see Figure 1). If the study participants did not explicitly withdraw their consent at the end of the T1 interview, the project staff called the individuals again after about 7 (T2) and 14 (T3) months. Figure 1 shows the participant flow. As expected for this population, participation rates were quite low: 3281 out of 4221 contacted individuals did not respond to the recruitment letter they received or declined participation.

The interviewers were trained in motivational interviewing such as that they responded with appreciation and encouragement, ensuring that interviewees provided valid answers that were uninfluenced by social desirability. Study participants were informed that the CATI was voluntary, that they could skip questions, or end the interview early if they felt too exhausted or overwhelmed. However, few individuals made use of this option: Most individuals adapted to the situation in general and were pleased to talk with the interviewer.

The CATIs took between 9.37 minutes (at T1; T2 = 10.87; T3 = 12.02) and 164.87 minutes (at T3; T1 = 129.33; T2 = 95.02). On average, the CATIs lasted *M* = 35.13 min at T1 (*SD* = 16.51; Median = 30.91; Mode = 22.33), *M* = 26.79 min at T2 (*SD* = 12.35; Median = 24.02; Mode = 18.38), and *M* = 48.41 min at T3 (*SD*=24.74; Median = 44.91; Mode = 39.27). Because the individuals were often not immediately reachable, many follow-up interviews were delayed and took place, on average, 34.7 (± 3.3) weeks after T1 for T2 and 67.1 (± 5.6) weeks after T1 for T3 (approximately 16.7 months).

All measures and the study design were piloted with the target population, and the procedure was approved by the Ethics Commission of the German Association of Psychology (Deutsche Gesellschaft für Psychologie; ID “SL012014_rev”). Reliability of measures was ensured by taking validated scales (e.g., [18]) and using qualitative pre-tests to check and adapt the items where necessary.

### 2.1. Measurements

All individuals who were insured with Deutsche Rentenversicherung Oldenburg-Bremen and were receiving TDP were invited to participate in T1. TDP status was reassessed at T2 and T3. To measure RTW-behavior, the following categories were formed: (a) TDP (full or part-time); (b) TDP converted to permanent DP; (c) TDP converted into old-age pension; (d) employed; (e) receipt of other benefits (unemployment pay).

*Income available to the household* was measured with the item “How well do you and your household manage to get along with the money available to you?“. The possible response options were “very badly” (1), “rather poorly” (2), ”rather well” (3), “very well” (4).

*Intention to return* to employment (RTW-intention) was assessed at T1, T2, and T3 with the following question: “Did you take any action to return to work after the end of your TDP independently of whether it would be full time or part time work, or a previous job or new job?”. The study participants were asked to indicate the most appropriate response, with the options (1) “No, I do not intend to return to work”, (2) “No, but I am considering it”, (3) “No, but I seriously intend to start”, (4) “Yes, and I have already prepared to return to work (e.g., talked to my employer; searched for a new job)”. These answers were dichotomized as “low” (1 or 2) or “high” (3 and 4).

*Subjective health* was measured with the question: “In general, how would you describe your health?” The answering options were (1) “poor”, (2) “fair”, (3) “good”, and (4) “very good or excellent”.

*RTW self-efficacy* was measured with two questions: “Now imagine you would return to work tomorrow. To what extent do you agree with the following statements, on a scale of 1 to 4?” The item was measured with a four-point scale, ranging from totally disagree (1) to totally agree (4). The statements were (A) “You face job-related problems without difficulties because you can always rely on your skills” and (B) “You will manage whatever happens in your working life”. 

*Social support* was measured with three questions: “Other people (e.g., friends, therapist, and family) have...” (A) “...encouraged you to return to work” (B) “...reminded you to contact your employer or write applications”, and (C) “...helped you to return to work”. The answering options were the same as those for self-efficacy expectations.

*Job autonomy and job demands* were introduced with the instruction "Now I’ll read you some statements. Please tell me on a scale of 1 to 4 how true each statement is for you with (1) meaning “Not at all true”, (2) “Barely true”, (3) “Mostly true” and (4) “Exactly true”. Job autonomy was measured by the question: “Did you have flexibility in your job?” Job Demands was measured by the question: “Did your work stress overwhelm you often?”.

*Outcome Expectations* (OE; motives) were measured using the potential reasons for returning to work, which were taken from previous research [18]. Participants who indicated that they did not intend to return to work, were asked to indicate which reasons might motivate them to return to work. The reasons assessed were (A) “...earn money”; (B) “...feeling needed”; (F) “...enjoyment of work” and (J) “...get a break from home/family”. Since the items had little to do with each other, no index was formed, and the individual OE were included separately in the analyses.

Furthermore, the *expectation regarding medical rehabilitation* was assessed by asking the study participants "What do you think? How much would medical rehabilitation help you to return to work at this point in time?” with the answering alternatives (1) “not at all”, (2) “a little”, (3) “somewhat”, and (4) “a great deal”.

The following *socio-demographic variables* related to TDP and employment were also assessed: Age, gender, body mass index (weight/(height*height)), household income, whether the pension was paid due to full or partial disability, number of children, vocational qualification (yes/no), and last employment (full-time, part-time, hourly or irregular, unemployment pay, homemaker/caretaking). Study participants were also asked whether they attended medical rehabilitation prior to receiving TDP (yes/no). They were also asked how long (in months) they had been receiving TDP. This was later categorized into 0–1 years, 1–2 years, 2–3 years, 4–6 years (2nd payment period), 7–9 years (3rd payment period), and >9 years.

### 2.2. Analyses

Where possible, inferential statistical analyses, such as χ² analyses and logistic regressions, were performed. Due to low cell frequencies, this was only possible for research question 3 and 4. Research questions 1 and 2 were therefore only investigated descriptively.

Dropout analyses revealed that there may have been a differential drop-out between T2 and T3. Those who did not respond at T3 were slightly more likely to report having more financial difficulties, were less motivated to return to work at T1, were less likely to be married or be partnered, and were more likely to be without children, *p* < 0.06. Because of the possibility of a distortion in representation between T2 and T3, the missing data were imputed employing the very conservative last observation carried forward (LOCF) method. The results of the original data are largely comparable to the imputed data.

## 3. Results

### 3.1. Research Question 1: What is the Likelihood of Returning to Work (RTW) when Receiving a Temporary Disability Pension (TDP) in Germany, and What are the Characteristics of those who Returned to Work?

Of the 453 respondents, 229 provided information on their employment status at T2 (out of a potential 232 insured persons who participated in T2, see Figure 1). For T3, 279 insured persons provided information on their employment status (out of a potential 286 insured persons who participated in T3, Figure 1).

The distributions of the work status of the 453 study participants at T2 and T3 are reported in Table 1. At T2, 16 individuals receiving TDP at T1 moved to permanent disability pension and two moved to old age pension. Of those individuals who received TDP at T2, more people transferred to permanent disability (*n* = 26) or to old-age pension (*n* = 1) than to RTW (*n* = 3) or other benefits (*n* = 4). One study participant managed to RTW at T2 and remained working at T3. In general, the few study participants with TDP at T1 returning to work at T2 and T3 are indicative of the low *likelihood of returning to work (RTW)*.

In order to examine the *characteristics of those who returned to work* (part 2 of research question 1, Table 2), descriptive analyses were used to compare those who were employed at T3 or were available to the labor market (i.e., registered unemployed), to those who were in TDP at T3, those who received permanent DP, and those who transitioned to old-age pension. Because cell sizes were too small, no inferential statistics were computed (Table 2).

*Gender* differences were investigated. Among the four individuals who managed to RTW, three were men; of the seven individuals who were available to the labor market but had not become actively employed, five were men. The percentage of men and women among those still in TDP, permanent disability pension or old-age pension at T3 was more equal, with the percentages ranging from 44.5% to 66.7%. In sum, more men than women returned to work or received other benefits at T3.

Regarding *age*, it appeared that those who managed to RTW were *M*_age_ = 46 years, those who remained in TDP *M*_age_ = 49.96 years, and those who were available to the labor market but had not become actively employed *M*_age_ = 52.43. Thus, those successfully returning to work were younger (Table 2).

The *body mass index (BMI)* of those study participants who started employment was *M*_BMI_ = 27.87 and was very similar to those who moved from TDP to permanent DP (*M*_BMI_ = 27.09) and to old-age pension (*M*_BMI_ = 29.23). Individuals who remained in TDP reported a higher BMI (*M*_BMI_ = 29.23). However, BMI was highest for those study participants moving from TDP to receiving other benefits, *M*_BMI_ = 32.98 (Table 2).

*Income available to the household* was rated on a four-point scale (from 1 to 4). While the full range was used by the study participants, the average was slightly above 2, corresponding to “rather poorly”. The mean was lowest in those moving to other benefits with *M*_available income_ = 1.83, followed by the individuals remaining in TDP with *M*_available income_ = 2.16, and those starting to work *M*_available income_ = 2.25. A relatively better rating was only given by the study participants moving into permanent disability pension (*M*_available income_ = 2.42) and to old-age pension (*M*_available income_ = 2.67).

*Time in TDP* prior to the first measurement point ranged widely. Those who returned to work received TDP for an average of 28 months, i.e., less than 2.5 years. This was shorter than for study participants remaining in TDP (*M*_time in TDP_ = 40.3), those who changed into permanent disability pension (*M*_time in TDP_ = 54.9) or to old-age pension (*M*_time in TDP_ = 57) or to other benefits (*M*_time in TDP_ = 73.29).

Study participants were asked whether they *received pension due to full or partially TDP* as an indicator of their disability severity. The ones who managed to return to work were equally likely to receive full or partial TDP (50%). Among the other four groups, there were more individuals who were fully disabled and were therefore receiving full TDP. Specifically, 86.7% of the ones remaining in TDP, and 100% of those changing to old-age pension were receiving full TDP. Further, 90.5% of those moving to permanent disability pension, and 80% of those receiving other benefits received pension due to full TDP.

*Subjective health* was best among those who had moved into old-age retirement (*M*_subj.health_ = 2.67), followed by those who returned to work (*M*_subj.health_ = 2.25). The health status of those who received other benefits was poorest (*M*_subj.health_ = 1.71). Also, the poor health of those who moved into permanent DP (*M*_subj.health_ = 1.85) validates that they were not just highly disabled but also subjectively very limited. The ones who remained in TDP had a rather poor health (*M*_subj.health_ = 1.99), compared to those who managed to return to work.

Having *children* can be a motivating factor to return to work: All individuals who moved into employment and 85.7% of those who moved to other benefits had children. In comparison, only 63%–71% of those remaining in TDP, moved into permanent DP, or moved into old-age pension had children (Table 2).

*Previous vocational training* was assessed in terms of education attained. All of the respondents who returned to work or received other benefits reported having vocational qualification. However, between 19.3% and 33.3% of those who remained in TDP, moved to permanent DP, or moved into old-age pensions reported having no previous vocational training.

Previous career paths in terms of the *last job* were explored. Whereas 75% of those who returned to work were full-time employees directly prior to the TDP, this was only the case for 71.4% of those who received other benefits; 57% of those who remained in TDP were working full-time in their last job. Only 33.3% of those who moved to old-age pension reported being employed fulltime prior to the TDP (further details see Table 2).

Around two out of every three study participants (62%) had *participated in a medical rehabilitation* during the last four years prior to the TDP. However, none of those who returned to work had done so. All other groups ranged between 62% (remaining in TDP) and 71.4% (receiving other benefits now).

### 3.2. Research Question 2: How does RTW-Behavior Relate to RTW-Intention?

Among those who returned to work at T3, 75% reported a high intention to RTW, significantly more than all other groups. Further, 71% of study participants who indicated being available to the labor market at T3 (i.e., receiving other benefits), and 58% of those still in TDP at T3 reported high intention to RTW at T1 (see Figure 2).

The majority of those in the other groups reported a low RTW-intention (see Figure 2). Because cell sizes were too small, no inferential statistics were computed. 

### 3.3. Research Question 3: Is There an Association between the Duration of TDP and the RTW-Intention?

To first test whether there is an association between the duration of TDP and the RTW-intention, only T1 data were used. Further, 63% of the study participants who had received TDP for a maximum of one year reported a high intention to RTW. This proportion became significantly smaller the longer participants were in TDP (Figure 3).

Only 52% of those who received TDP between four and six years (second approval period) reported a high RTW-intention. Of those who received TDP between seven and nine years (third approval period) only 40% reported a high RTW-intention, and 30% of those who received TDP for 9 or more years reported a high RTW-intention (Figure 3). This correlation was statistically significant with χ² (*N* = 441; *df* = 5) = 11.73; *p* = 0.04, or Kendall’s Tau *b* = -0.12, *p* = 0.01.

The finding was replicated among those study participants who continued to receive TDP at T3 (less than 1 year: 64% high intention; 1–2 years 49%; 3–4 years 61%; second approval period 53%; third approval period 43% and more than 9 years 24%; χ² (*N* = 389; *df* = 5) = 13.22; *p* = 0.02 or Kendall’s Tau *b* = −0.11, *p* = 0.01). In summary, the proportion of those study participants with low RTW-intentions seemed to grow with the duration in TDP, or put differently, the longer the individuals were in TDP, the lower their intention to return to work.

### 3.4. Research Question 4: What Socio-Demographic and Psychological Factors Predict a High RTW-Intention?

Logistic regressions were calculated to determine which factors were related to a high RTW-intention. Intention at T1, as the dependent variable, was first predicted in two steps. In the *first step*, socio-demographic factors (gender, age, BMI, time in the TDP prior to T1, and income in the form of a subjective assessment on how well the respondents cope with available money), personal resources (social-cognitive variables related to employment such as RTW self-efficacy and social support), as well as job autonomy and the job demand, were analyzed.

When predicting RTW-intention cross-sectionally at T1, only self-efficacy was significant after controlling for socio-demographic variables; respondents who were younger (OR = 0.93), had been receiving TDP for a shorter period of time (OR = 0.78), and who had been struggling financially (OR = 0.75) had a higher RTW-intention. RTW-intention was higher with a higher self-efficacy expectation (OR = 1.26; Table 3). These variables explained 20% of the RTW-intention at T1.

Subjective health, OE, and the expectation that medical rehabilitation might be helpful to RTW were all added in the *second step*. While age and time in TDP remained statistically significant, the OE to earn money (OR = 1.99), the OE to feel needed (OR = 1.71), and the OE to get a break from home/family (OR = 1.41) were significantly correlated with RTW-intention (but not the OE of enjoyment of work and subjective health). The expectation that medical rehabilitation would help to RTW was also significant (OR = 1.55) for RTW-intention. In total, 37% of the variance in RTW-intention at T1 was explained.

In order to also predict the change in RTW-intention from T1 to T3, this second step was repeated with RTW-intention at T3, after controlling for RTW-intention at T1 in a *third step*. While this model could explain 32% of the T3-variance, only age (OR = 0.91) and subjective health (OR = 2.37) were significant.

In summary, a high intention to return to work at T1 and T3 was more likely the younger the individuals were. The intention at T1 was additionally predicted by a shorter time in TDP and some outcome expectancies as well as the expectation regarding medical rehabilitation. However, a higher intention at T3 was instead more likely if subjective health was better while all other variables were no longer statistically significant.

## 4. Discussion

In this longitudinal survey, 453 individuals who received temporary disability pension (TDP) at the first measurement point were assessed over 17 months. The findings confirm previous results [27,28] in that only a small proportion of individuals in TDP were able to return to work or became available for the labor market. Only 2.4% of individuals in this study were able to return to work or became available to the labor market, much lower than comparable data. Germany reports around 8% [28] of disability pensioners RTW, and those with mental disorders at around 6% RTW [27]; whereas Norway reports 11% RTW [8]. However, our findings are comparable with findings from Sweden, which show a 0%–1% RTW rate [29]. Evidently, there is a need for effective approaches to help individuals in TDP to RTW. Particularly, individuals with TDP whose health status has recovered should receive support in regaining social participation and RTW by pointing out the advantages of working life (outcome expectancies). Those with suboptimal health should also be adequately supported, especially through facilitating medical rehabilitation. Also, occupational health promotion should help those employees with persistent functional limitations who are at risk for DP with the long-term management of their condition [2,3,16] and two overcoming barriers (that) may hinder their full and effective participation in society [9] (pp. 4).

Our data indicate that RTW-intention is clearly related to RTW-behavior. This supports previous findings in individuals on sick leave but not yet in disability pension [1,13,18,19] and extends these with data from individuals in TDP. This shows that it is important to continue to increase and maintain support for individuals with TDP to build an intention to return to (previous) work, the labor market, or search for a job [16].

Important predictors of this intention were examined based on the *Job Demand Resource Model* [20] and it was shown that self-efficacy and OE were imperative. To be precise, higher OEs to earn money, to feel needed, and to get a break from family/home were associated cross-sectionally with a higher RTW-intention, but the OE of enjoying work was not statistically significant. Also, job demands and job resources were not significantly correlated with RTW-intention. This contradicts findings from the literature [5] and suggests that the study participants already felt disconnected to the job and perceived these aspects, especially self-efficacy, to be unimportant as personal resources. However, this finding may also be attributed to the sample, which might be rather selective, e.g., poorly fit for their last job and therefore not enjoying work. On one hand, personal resources such as self-efficacy and outcome-expectancies need to be restored or developed in interventions [14]. On the other hand, it must also be critically noted that the study participants may have a “realistic” idea of their chances of returning to the labor market, considering their physical and mental limitations.

RTW is certainly not easy, especially for those with fewer skills. TDP seems to be a “reservoir” of people who are rather difficult to employ due to physical or mental health reasons. It could be that these people already had no expectations of regular employment prior their entry into TDP, as was found in the employment histories before TDP. In order to help people in TDP, their self-efficacy expectations to RTW despite their age and their disabilities could be addressed through, for example, staged work trials or internships in which mastery skills are attained. These trainings are already offered in the field of work-related medical rehabilitation [2,3]. However, these treatments need to be delivered more appropriately to and actually accepted by the individuals in TDP.

Medical rehabilitation and occupational health promotion should be tailored more strongly to the prevention of DP and to the special characteristics of the individuals in TDP [11,16]. Analyses of the importance of duration in TDP show that measures should be taken as early as possible. The research question on the significance of duration in reduced earning capacity was examined in two analyses. The frequency analysis showed a significant negative correlation between time in TDP and RTW-intention. When age and duration were simultaneously integrated in a regression analysis, both were significant cross-sectional predictors of RTW. In other words, although the two variables both reflect time aspects, they have discriminant validity. Aging seems partly distinct from simply staying in DP longer. This is probably also correlated with the subjective perception of one’s health and disabilities.

Certainly, the individuals with TDP have already received various forms of support, e.g., in the form of information material by pension insurance companies or medical treatment by their medical doctor or other medical personnel. However, research has shown that it is difficult to motivate employees who feel unable to work towards RTW [30,31]. In an intervention [31], workers were supported in their motivation, objectives, and concrete action planning, but this did not lead to an increase in work attendance compared to a control group, nor to an improvement in employment or gainful employment after one year. From this, it can be concluded that support focusing exclusively on providing information and increasing motivation is not enough. Structural support measures seem to be necessary within jobs so that despite physical or mental limitations, positive self-efficacy-relevant experiences can be made and disability overcome [10,12,14]. 

In the prediction of RTW-intention, three steps were taken to sequentially analyze the significance of variables and to understand intention, both cross-sectionally and in terms of their changes over time. In the cross-sections, those who have a higher intention to RTW were younger, were in TDP for a shorter time, reported less available household income, reported more self-efficacy, higher OE to earn money, felt needed, worked to get a break from the family/home, and had higher expectations for medical rehabilitation than those who have a lower RTW-intention.

Longitudinally, only age and subjective health predicted higher RTW-intentions at T3, underlining the importance of good health, and thereby replicating previous findings [12]. In order to help individuals in TDP to improve their subjective health status, lifestyle-related interventions such as physical activity, weight reduction, smoking cessation, relaxation procedures, etc., can be useful [10,14]. While medical rehabilitation is already aimed at this, the population of temporary disability pensioners is certainly underserved in terms of being admitted to rehabilitation. This fact, along with our finding that the expectation regarding medical rehabilitation seemed to support higher RTW-intention, points out that proactively advertising and fostering access to medical rehabilitation services in this population seems central to enable RTW. Temporary disability pensioners need resources that can assist them in improving their health and coping with job demands. Interestingly, subjective health was best among those who had moved into old-age retirement. It may be speculated that without the external pressure to RTW, individuals felt less affected by their disabled health status; or being approved for transition into retirement actually helped them to recover.

Due to low cell sizes, some inferential analyses could not be conducted, and the results should therefore be interpreted with caution. Furthermore, the applied last observation carried forward (LOCF) imputation method may not be the most appropriate, and validity of the imputed data needs to be considered. Also, in general, low response rates indicated that this was a difficult sample to study. Maybe more intense forms of assessments such as face-to-face interviews at the homes of the individuals would be more appropriate for achieving a more representative picture of this group. However, considering this group is so delicate, the response rate and the attained sample size is acceptable when compared to the few other studies [e.g., 8]. Furthermore, the longitudinal nature of this study is a strength, and is rather unique for such investigations. Accordingly, some moderate recommendations can be derived from the study. (1) Offering support at an early stage and designing it to fit individual needs, especially taking their subjective health into account. (2) Counteracting age stereotypes to strengthen RTW-intention (through incentives if necessary). (3) Strengthening self-efficacy expectations and enabling mastery experiences. (4) Outlining to individuals the financial incentives and other social benefits that relate to OE they can gain from RTW, especially in the beginning of their TDP. (5) Offering medical rehabilitation in conjunction with work-related content to TDP to support the everyday functioning and coping with health conditions. In the long term, other expectations and experiences should be added. (6) Those whose risk profile reveals a potential to change from TDP to being available to the labor market should be supported by occupational health services and in “career building” in such a way that they also have a chance of RTW and that there is no shift from one state to another without RTW. In general, RTW rates need to be greatly improved not only to realize more potential RTW but to especially also enable each individual to participate socially in society, thereby affirming a human right [9].

## 5. Conclusions

Medical and vocational rehabilitation should be offered to a larger amount of disability pensioners in order to try to improve currently suboptimal RTW rates. Because RTW appeared to be related to previous RTW-intention, this should be considered when improving medical and vocational rehabilitation.RTW is more likely in the individuals who are younger and in TDP for a shorter period of time. However, these two factors appear to be rather independent of one another and rehabilitation should be offered as soon as possible, especially to younger people.Those who are younger and have been in TDP for less time have a higher RTW-intention. More effort has to be put into rehabilitation to help disability pensioners, especially those in older age, to increase their intention to RTW.Self-efficacy and OE to earn money, feel needed, and get a break from their family/home environment are important in the development of RTW-intentions. This should be addressed in rehabilitation with information and behavior change interventions.Good health is central to RTW-intention. (Work-related) medical rehabilitation can facilitate overcoming disability challenges and improving health status.

## Figures and Tables

**Figure 1 ijerph-17-00238-f001:**
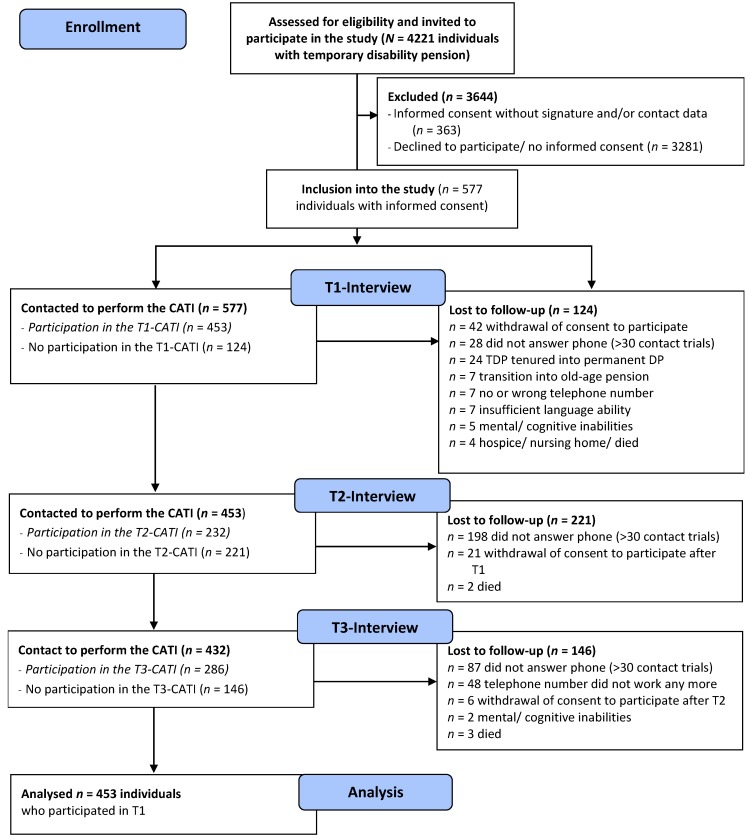
Consort flow diagram.

**Figure 2 ijerph-17-00238-f002:**
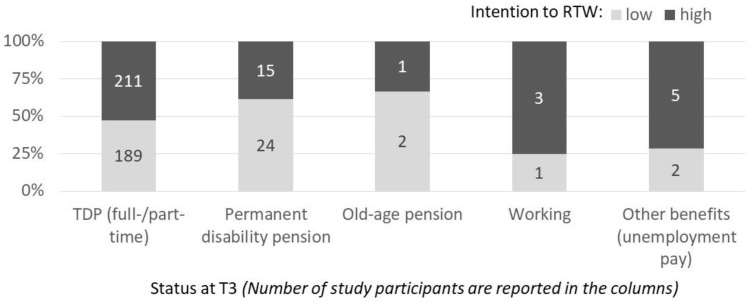
Number of study participants in the different status groups at T3 and their RTW-intention at T1 (high/low). *Note.* Intention to RTW = intention to return to work at T1 dichotomized.

**Figure 3 ijerph-17-00238-f003:**
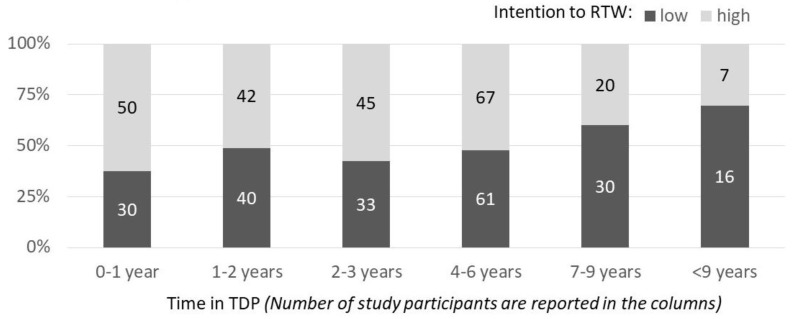
Number of individuals with high (light grey) and low (dark grey) RTW-intention who receive TDP for different time frames at T1. *Note*. TDP = temporary disability pension; intention to RTW = intention to return to work at T1 dichotomized.

**Table 1 ijerph-17-00238-t001:** Work status of study participants at T2 and T3: Whether they maintained in temporary disability pension (TDP), moved to permanent disability pension, old-age pension, employment, or other benefits with imputed numbers (last-observation-carried-forward method) as well as participants who actually provided data (in brackets).

*T1:* All Study Participants Were in TDP		*T3*					Total
		TDP (Full-/Part-Time)	Permanent Disability Pension	Old-age Pension	Working	Other Benefits (Unemploy-Ment Pay)	
***T2*** **TDP (full-/part-time)**	number	396 (149)	26 (8)	1 (1)	3 (1)	4 (2)	430 (161)
	%	87.4% (86.1%)	5.7% (4.6%)	0.2% (0.6%)	0.7% (0.6%)	0.9% (1.2%)	94.9% (93.1%)
**Permanent disability pension**	number	3 (3)	13 (5)	0 (0)	0 (0)	0 (0)	16 (8)
	%	0.7% (1.7%)	2.9% (2.9%)	0% (0%)	0% (0%)	0% (0%)	3.5% (4.6%)
**Old-age pension**	number	0 (0)	0 (0)	2 (2)	0 (0)	0 (0)	2 (2)
	%	0% (0%)	0% (0%)	0.4% (1.2%)	0% (0%)	0% (0%)	0.4% (1.2%)
**Working**	number	0 (0)	0 (0)	0 (0)	1 (0)	0 (0)	1 (0)
	%	0% (0%)	0% (0%)	0% (0%)	0.2% (0)	0% (0%)	0.2% (0%)
**Other benefits (unemployment pay)**	number	1 (1)	0 (0)	0 (0)	0 (0)	3 (1)	4 (2)
	%	0.2% (0.6%)	0% (0%)	0% (0%)	0% (0%)	0.7% (1.7%)	0.9% (1.2%)
**Total**	number	400 (153)	39 (13)	3 (3)	4 (1)	7 (3)	453 (173)
	%	88.3% (88.4%)	8.6% (7.5%)	0.7% (1.7%)	0.9% (0.6%)	1.5% (1.7%)	100.0% (100%)

**Table 2 ijerph-17-00238-t002:** Characteristics of return-to-work (RTW) status at T3 by variables at T1 (age, BMI, etc.): Mean values, standard deviation in brackets, range, or percentages.

Characteristics	TDP (Full-/Part-Time)	Permanent Disability Pension	Old-age Pension	Working	Other Benefits (Unemploy-Ment Pay)	Total
	*N* ≤ 400	*N* ≤ 39	*N* ≤ 3	*N* ≤ 4	*N* ≤ 7	*N* ≤ 453
Gender (males)	178 (44.5%)	22 (56.4%)	2 (66.7%)	3 (75%)	5 (71.4%)	210 (46.4%)
Age (in years)	49.96 (7.89) 27–63	53.85 (6.58) 35–62	61.67 (3.06) 59–65	46.00 (12.73) 28–58	52.43 (6.27) 41–59	50.38 (7.90) 27–65
Body Mass Index	29.23 (7.45) 16.53–60.14	27.09 (5.14) 17.32–38.70	27.51 (4.97) 23.99–31.02	27.87 (3.14) 24.93–31.56	32.98 (8.40) 22.59–47.02	29.08 (7.28) 16.53–60.14
Income available to the household	2.16 (0.89) 1–4	2.42 (0.64) 1–4	2.67 (0.58) 2–3	2.25 (0.50) 2–3	1.83 (0.75) 1–3	2.18 (0.87) 1–4
Time in TDP (months)	40.30 (31.01) 1.00-170	54.90 (49.70) 6.00-277.00	57.00 (51.00) 6.00-108.00	28.25 (29.19) 12.00-72.00	73.29 (73.71) 9.00-218.00	42.08 (34.40) 1.00-277.00
Pension due to full TDP	86.7%	90.5%	100.0%	50.0%	80.0%	86.8%
Subjective health	1.99 (0.69) 1–4	1.85 (0.67) 1–4	2.67 (1.16) 2–4	2.25 (0.50) 2–3	1.71 (0.76) 1–3	1.98 (0.69) 1–4
Children	281 (70.6%)	24 (63.2%)	2 (66.7%)	4 (100%)	6 (85.7%)	317 (70.4%)
No vocational qualification	77 (19.3%)	8 (20.5%)	1 (33.3%)	0 (0.0%)	0 (0.0%)	86 (19.1%)
Last job was……Full time	57.0%	56.4%	33.3%	75.0%	71.4%	57.2%
…Part-time	20.3%	12.8%	33.3%	25.0%	0.0%	19.4%
…By the hours/unregularly	4.1%	0.0%	0.0%	0.0%	0.0%	3.3%
…Unemploy-ment pay/	12.3%	17.9%	33.3%	0.0%	28.6%	12.3%
…Homemaker/Caretaking ^1^	2.8%	5.1%	0.0%	0.0%	0.0%	2.9%
Participation in medical rehabilitation	248 (62.0%)	26 (66.7%)	2 (66.7%)	0 (0.0%)	5 (71.4%)	281 (62.0%)

^1^ = Missing percentages: Training/continuing education/studying or miscellaneous.

**Table 3 ijerph-17-00238-t003:** Prediction of RTW-intention (low vs. high) with binary-logistic regression analyses.

Predictor Variables	Step 1 Prediction T1 RTW-Intention ^1^			Step 2 Prediction T1 RTW-Intention ^1^			Step 3 Prediction T3 RTW-Intention Change ^2^		
	OR	95% CI for Lower	95% CI for Upper	OR	95% CI for Lower	95% CI for Upper	OR	95% CI for Lower	95% CI for Upper
Sex	1.259	0.760	2.086	0.967	0.549	1.705	0.763	0.315	1.848
Age	0.930**	0.899	0.962	0.947**	0.912	0.983	0.909**	0.857	0.964
Body Mass Index	1.017	0.982	1.054	1.015	0.975	1.056	1.015	0.960	1.073
Time in TDP	0.782*	0.645	0.947	0.790*	0.640	0.977	0.825	0.600	1.135
Income available to the household	0.753*	0.566	0.999	0.951	0.678	1.335	1.552	0.947	2.544
Self-efficacy	1.261**	1.106	1.438	1.139	0.981	1.322	1.073	0.850	1.354
Social support	1.062	0.970	1.163	1.018	0.920	1.126	0.962	0.840	1.101
Job autonomy	1.108	0.881	1.394	1.069	0.829	1.379	1.213	0.817	1.802
Job demand	1.044	0.822	1.326	1.023	0.785	1.334	0.923	.621	1.371
OE to earn money				1.987**	1.295	3.048	1.673	0.874	3.201
OE to feel needed				1.710**	1.143	2.558	1.174	0.554	2.490
OE enjoyment of work				0.983	0.585	1.652	1.505	0.563	4.024
OE to get a break				1.409*	1.058	1.876	1.393	0.886	2.189
Subjective health				0.990	0.643	1.526	2.369*	1.182	4.749
Expectation of medical rehabilitation				1.547**	1.186	2.017	1.330	0.918	1.926
T1 RTW-Intention							1.742	0.698	4.347
*R* ^2^	0.20			0.37			0.32		

*Note*: Findings are mainly based on those that remain in TDP, i.e., results were replicated with this subsample; ^1^ = RTW-intention low vs. high at T1; ^2^ = RTW-intention low vs. high at T3 controlled for T1 RTW-intention. OE = Outcome Expectancies; RTW = Return to work; * = *p* ≤.05; ** = *p* ≤.01.

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
