# Peer review of "Temporary Disability Pension, RTW-Intentions, and RTW-Behavior: Expectations and Experiences of Disability Pensioners over 17 Months"

_ijerph, 2019, doi:10.3390/ijerph17010238_

Round 1
Reviewer 1 Report
My main criticism concerns the low response rate of the survey; only 453 participants out of the study population of 4 221 were interviewed in T1, and out of these 453 only 232 in T2. For T3 interview 432 persons were contacted, out of whom 286 were interviewed. Further, the authors have used imputation method to estimate the missing data, and thus keep the 453 T1 participants in the analysis. The imputation method has not been described, and the authors only mention that "The results of the original data are largely comparable to the imputed data". Also, in the discussion section the possible effect of this imputation to the results is not discussed at all.
Then when you look to Table 1 you will find out that the cases in outcome variables (employment or other benefits) are very few 1 and 4 in T2; 4 and 7 in T3 out of 453 participants. Are this cases based on imputation, and if, how was this done? Based on flow diagram fig 1. the data was only available from 232 interviewed in T2 and 286 interviewed in T3.
I would suggest that even though the number of participants drops almost to half, the data of 232 (T2) and 286 (T3) participants should be used for the analysis.
In figure 3 the n:s for each bar should be presented (working n = 4; other benefits n = 7). Is it worth counting percentages with decimals for such a few cases?
In table 3 no reference groups are shown for each category; there are also *, **marks without any explanation of what these mean.
In line 71 it is mentioned, that the baseline data (T1) of this study have been published elsewhere, but no reference is given. However in answering the research question 3 only data from T1 was used; it would have been useful to see what was already published earlier of the same data.
Author Response
Author's Reply to the Review Report (Reviewer 1)
Reviewer Comment: My main criticism concerns the low response rate of the survey; only 453 participants out of the study population of 4 221 were interviewed in T1, and out of these 453 only 232 in T2. For T3 interview 432 persons were contacted, out of whom 286 were interviewed.
Author's Reply: We are very thankful for this comment and the chance to improve our manuscript as we totally agree: The low response rate is a problem but also an indicator of the challenges of the individuals facing long-lasting disabilities preventing them from working and social participation, the need to return to work as well as to cope with social stigma. Accordingly, we included this aspect more explicitly into the introduction and discussion and hope that the readers now understand better why the response rate is comparatively acceptable. In detail, we state
“This dearth of studies may partially be due to the fact that this is a difficult population to study—RTW rates are usually quite low and disability pensioners may feel stigmatized and reluctant to talk about their status.” (lines 36-38)
“…small sample sizes and low response rates are not only a problem but also indicative of the challenges experienced by individuals facing long term disabilities: preclusion from work and employment in specific, and participation in society in general – which are human rights [9]..” (lines 44-47).
“Figure 1 shows the participant flow. As it is to be expected for this population, participation rates were quite low: 3,281 out of 4,221 contacted individuals did not respond to the recruitment letter they received, or declined participation.” (lines 80-83).
and
„ Also, in general, low response rates indicated that this was a difficult sample to study. Maybe more intense forms of assessments such as face-to-face interviews at the homes of the individuals would be more appropriate for achieving a more representative picture of this group. However, considering this group is so delicate, the response rate and the attained sample size is acceptable when compared to the few other studies [e.g., 8]. Furthermore, the longitudinal nature of this study is a strength, and is rather unique for such investigations.“ (lines 348-353).
Reviewer Comment: Further, the authors have used imputation method to estimate the missing data, and thus keep the 453 T1 participants in the analysis. The imputation method has not been described, and the authors only mention that "The results of the original data are largely comparable to the imputed data". Also, in the discussion section the possible effect of this imputation to the results is not discussed at all.
Author's Reply: Thank you very much for the feedback and asking us to describe the imputation method and the actual data further. We have done so by moving the section on the imputation procedure from the result section to the method section and expanding this (the section now reads “Dropout analyses revealed that there may have been a differential drop-out between T2 and T3. Those who did not respond at T3 were slightly more likely to report having more financial difficulties, were less motivated to return to work at T1, were less likely to be married or be partnered, and were more likely to be without children, p<.06. Because of the possibility of a distortion in representation between T2 and T3, the missing data were imputed employing the very conservative last observation carried forward (LOCF) method. The results of the original data are largely comparable to the imputed data.” (lines 152-158).
Furthermore, we added to the heading of table 1 so it now reads “Table 1. Work status of study participants at T2 and T3: Whether they maintained TDP, moved to permanent DP, old-age pension, employment or other benefits with imputed numbers (last-observation-carried-forward method) as well as participants who actually provided data (in brackets).” (lines 175-178).
Additionally, to complete the discussion section reviewing the possible effect of this imputation to the results we added the sentence “Furthermore, the applied last observation carried forward (LOCF) imputation method may not be the most appropriate, and validity of the imputed data needs to be considered.“ (lines 346-348).
Reviewer Comment: Then when you look to Table 1 you will find out that the cases in outcome variables (employment or other benefits) are very few 1 and 4 in T2; 4 and 7 in T3 out of 453 participants. Are this cases based on imputation, and if, how was this done? Based on flow diagram fig 1. the data was only available from 232 interviewed in T2 and 286 interviewed in T3.
Author's Reply: Thank you for this comment and we appreciate the option to add the information with the non-imputed, original data to Table 1. We now explain that we used participants’ last available status specification in this table. This imputation procedure is the last observation carried forward (LOCF) method and assumes that individuals who do not participate in a measurement point fell back into the state before and are therefore not willing to participate as they expect to only report changes but no maintenance. This is a procedure which was also applied in similar studies, such as:
Hall, I., Parkes, C., Samuels, S., & Hassiotis, A. (2006). Working across boundaries: clinical outcomes for an integrated mental health service for people with intellectual disabilities. Journal of Intellectual Disability Research, 50(8), 598-607.
Eberhardt, K., Kapetanovic, M. C., Lindqvist, E., Nilsson, J. Å., Saxne, T., & Geborek, P. P. (2013). FRI0082 Development of impairment, disability, radiological damage, comorbidity and mortality over 20 years in rheumatoid arthritis patients. Annals of the Rheumatic Diseases, 71(Suppl 3), 337-337.
Katharina Büsch, Simone A. da Silva, Michelle Holton, Fabiana M. Rabacow, Hamed Khalili, Jonas F. Ludvigsson, Sick leave and disability pension in inflammatory bowel disease: A systematic review, Journal of Crohn's and Colitis, Volume 8, Issue 11, 1 November 2014, Pages 1362–1377,
Please note, in Figure 1, we did not include any imputed data as this is the report of participant flow with original data only.
Reviewer Comment: I would suggest that even though the number of participants drops almost to half, the data of 232 (T2) and 286 (T3) participants should be used for the analysis.
Author's Reply: Thank you for this suggestion which we carefully considered. However, we maintained the results as before as a higher statistical power increases reliability of findings as well. We believe this to be acceptable with the addition of a cautionary sentence to the discussion: “Due to low cell sizes, some inferential analyses could not be conducted, and the results should therefore be interpreted with caution. Furthermore, the applied last observation carried forward (LOCF) imputation method may not be the most appropriate, and validity of the imputed data needs to be considered.” (lines 345-348).
Reviewer Comment: In figure 3 the n:s for each bar should be presented (working n = 4; other benefits n = 7). Is it worth counting percentages with decimals for such a few cases?
Author's Reply: Thank you very much for this suggestion which certainly facilitates interpretation, we have added the requested numbers Figure 2 and 3.
Reviewer Comment: In table 3 no reference groups are shown for each category; there are also *, **marks without any explanation of what these mean.
Author's Reply: We are very sorry for this omission and now provide the information and added a note to explain *, ** (* p≤.05; ** = p≤.01.).
Reviewer Comment: In line 71 it is mentioned, that the baseline data (T1) of this study have been published elsewhere, but no reference is given. However in answering the research question 3 only data from T1 was used; it would have been useful to see what was already published earlier of the same data.
Author's Reply: The findings provided in research question 3 have not been published before. We now disclose the actual reference which we originally have masked to enable an anonymous review process.
Zschucke, E.; Hessel, A.; Lippke, S. Befristete Erwerbsminderungsrente aus Sicht der Betroffenen: subjektiver Gesundheitszustand, Rehabilitationserfahrungen und Pläne zur Rückkehr ins Erwerbsleben [Temporary disability pension from the perspective of the individual: self-reported physical and mental health, medical rehabilitation, and return to work plans]. Die Rehabilitation. 2016, 55(04), 223-229; DOI: 10.1055/s-0042-109574.
Reviewer 2 Report
Thank you for the opportunity to review this manuscript. The topic has universal application for most all societies who struggle with a work force that is unable/unwilling/unmotivated to return to work following a disruption of some kind.
There appeared to be a plethora of information presented; which arguably was at times, overwhelming to absorb. Addressing the analysis question by question helped, yet within each question, it was hard to process. The issue is not the analytical techniques, but the way in which it was presented and even more so, how the questions were developed.
I did not see any discussion about the piloting process- how were the questions developed determined to be reliable? This is a concerning omission.
The "intention to return to employment" the neutral (?) response "No, but I seriously intend to start" does not seem to make sense. Perhaps it does not translate from German to English adequately, but it none the less, does not appear to be cogent.
How long did the survey questions take (average) to complete? What consideration was made for participants who may be overwhelmed by the entire loss of labor and return to work issue and challenged to respond? A more pragmatic discussion would have been nice to include.
The conclusions could have been stronger and more definitive as to how solutions could be implemented and more so, how those who are on TDP could be motivated to RTW. I feel this section would have been strengthened by a more robust and thoughtful and more humanistic exploration of solutions rather than blanket statements to address the issue.
Thank you.
Author Response
Author's Reply to the Review Report (Reviewer 2)
Reviewer Comment: Thank you for the opportunity to review this manuscript. The topic has universal application for most all societies who struggle with a work force that is unable/unwilling/unmotivated to return to work following a disruption of some kind.
Author's Reply: Thank you very much for this positive feedback.
Reviewer Comment: There appeared to be a plethora of information presented; which arguably was at times, overwhelming to absorb. Addressing the analysis question by question helped, yet within each question, it was hard to process. The issue is not the analytical techniques, but the way in which it was presented and even more so, how the questions were developed.
Author's Reply: We have now tried to improve the flow of the manuscript by putting the key aspects in italics such as “One study participant managed RTW at T2 and remained working at T3. In general, the few study participants with TDP at T1 returning to work at T2 and T3 are indicating the low likelihood of returning to work (RTW).“ (lines 171-173) and „In order to examine the characteristics of those who returned to work…“ (line 179).
Additionally, we have tried to trim the text and to better present the development of the research questions and the description of the results.
Reviewer Comment: I did not see any discussion about the piloting process- how were the questions developed determined to be reliable? This is a concerning omission.
Author's Reply: Thank you very much pointing this out. We did not report this because we intended to keep the report as short as possible. Of course, we piloted the measures and the procedure and report this now as follows:
“All measures and the study design were piloted with the target population, and the procedure was approved by the Ethics Commission of the German Association of Psychology (Deutsche Gesellschaft für Psychologie; ID „SL012014_rev”). Reliability of measures was ensured by taking validated scales (e.g., [18]) and using qualitative pre-tests to check and adapt the items where necessary.” (lines 96-100).
Reviewer Comment: The "intention to return to employment" the neutral (?) response "No, but I seriously intend to start" does not seem to make sense. Perhaps it does not translate from German to English adequately, but it none the less, does not appear to be cogent.
Author's Reply: Thank you very much for detecting this translation error! The stem of the question should actually read (and is now corrected) “Did you take any action to return to work after the end of your TDP independently of whether it would be full time or part time work, or a previous job or new job?” The study participants were asked to indicate the most appropriate response, with the options (1) „No, I do not intend to return to work”, (2) „No, but I am considering it”, (3) „No, but I seriously intend to start”, (4) „Yes, and I have already prepared to return to work (e.g., talked to my employer; searched for a new job)”. These answers were dichotomized as „low” (1 or 2) or „high” (3 and 4).” (lines 108-113).
Reviewer Comment: How long did the survey questions take (average) to complete?
Author's Reply: Thank you asking such an important question and we are happy to add the regarding information: “The CATIs took between 9.37 minutes (at T1; T2=10.87; T3=12.02) and 164.87 minutes (at T3; T1=129.33; T2=95.02). On average, the CATIs lasted M=35.13 min at T1 (SD=16.51; Median=30.91; Mode=22.33), M=26.79 min at T2 (SD=12.35; Median=24.02; Mode=18.38) and M=48.41 min at T3 (SD=24.74; Median=44.91; Modus=39.27). Because the individuals were often not immediately reachable, many follow-up interviews were delayed and took place on average 34.7 (± 3.3) weeks after T1 for T2, and 67.1 (± 5.6) weeks after T1 for T3 (approx. 16.7 months).” (lines 90-95).
Reviewer Comment: What consideration was made for participants who may be overwhelmed by the entire loss of labor and return to work issue and challenged to respond? A more pragmatic discussion would have been nice to include.
Author's Reply: We appreciate the opportunity to explain our procedure further and did so in the manuscript by adding “The interviewers were trained in motivational interviewing such as that they responded with appreciation and encouragement, ensuring that interviewees provided valid answers that were uninfluenced by social desirability. Study participants were informed that the CATI was voluntary, that they could skip questions or end the interview early if they felt too exhausted or overwhelmed. However, few individuals made use of this option: Most individuals adapted to the situation in general and were pleased to talk with the interviewer.” (lines 84-89).
Reviewer Comment: The conclusions could have been stronger and more definitive as to how solutions could be implemented and more so, how those who are on TDP could be motivated to RTW. I feel this section would have been strengthened by a more robust and thoughtful and more humanistic exploration of solutions rather than blanket statements to address the issue.
Author's Reply: Thank you very much, we revised the discussion and conclusion section, e.g., by now stating “In general, RTW rates need to be greatly improved not only to realize more potential RTW but to especially also enable each individual to participate socially in society, thereby affirming a human right [9].” (lines 364-366).
We hope the reviewer agrees with us that this revised version of the discussion and conclusion is now more robust and thoughtful. We believe the manuscript improved by the more humanistic exploration of solutions and the omission of blanket statements.
Round 2
Reviewer 1 Report
The authors have addressed adequately the comments I had for the previous version of the manuscript. Introduction, material and methods as well as the results sections have been improved with additional information. Also, the English language is now improved. In the discussion section the authors have pointed out the weaknesses of the statistical analyses due to the need for imputation of the missing data.
Author Response
Author's Reply: Thank you very much for this positive feedback.
Reviewer 2 Report
Thank you for resubmitting a revised manuscript. It is evident that the authors put forth a great deal of effort in doing so. Most commendable is the vastly improved discussion section, which is more comprehensive.
The results section remains challenging to read and process secondary to the amount of information that is shared. A further revision, working closely with a skilled copy editor could be helpful for the authors to produce a more highly readable manuscript on a topic that is important and timely- RTW.
Thank you for the opportunity to review this manuscript.
Author Response
Reviewer Comment: Thank you for resubmitting a revised manuscript. It is evident that the authors put forth a great deal of effort in doing so. Most commendable is the vastly improved discussion section, which is more comprehensive.
Author's Reply: We very much appreciate this encouraging feedback.
Reviewer Comment: The results section remains challenging to read and process secondary to the amount of information that is shared. A further revision, working closely with a skilled copy editor could be helpful for the authors to produce a more highly readable manuscript on a topic that is important and timely- RTW.
Author's Reply: Thank you very much for the opportunity to improve readability. We reworked the whole results section and also consulted again with our skilled copy-editing assistant. Many revisions were realized (please see the manuscript with all revisions highlighted attached). We are sure that communication significantly improved. We hope the reviewer and the editor agree and hope to see the manuscript accepted for publication soon.